# A Compensation Method for the Geomagnetic Measurement Error of an Underwater Ship-Borne Magnetometer Based on Constrained Total Least Squares

**DOI:** 10.3390/s24113478

**Published:** 2024-05-28

**Authors:** Yude Tong, Xiaoying Huang, Yongbing Chen, Wenkui Li, Feng Zha

**Affiliations:** 1College of Electrical Engineering, Naval University of Engineering, Wuhan 430033, China; tongyude@126.com (Y.T.); hgcyb@163.com (Y.C.); lwk404@tom.com (W.L.); zha_feng@126.com (F.Z.); 2Department of Operational Research and Programing, Naval University of Engineering, Wuhan 430033, China

**Keywords:** constrained total least squares, geomagnetic measurement, compensation method, ship-borne magnetometer

## Abstract

When magnetic matching aided navigation is applied to an underwater vehicle, the magnetometer must be installed inside the vehicle, considering the navigation safety and concealment of the underwater vehicle. Then, the interference magnetic field will seriously affect the accuracy of geomagnetic field measurement, which directly affects the accuracy of geomagnetic matching aided navigation. Therefore, improving the accuracy of geomagnetic measurements inside the vehicle through error compensation has become one of the most difficult problems that requires an urgent solution in geomagnetic matching aided navigation. In order to solve this problem, this paper establishes the calculation model of the internal magnetic field of the underwater vehicle and the geomagnetic measurement error model of the ship-borne magnetometer. Then, a compensation method for the geomagnetic measurement error of the ship-borne magnetometer, based on the constrained total least square method, is proposed. To verify the effectiveness of the method proposed in this paper, a simulation experiment of geomagnetic measurement and compensation of a ship-borne three-axis magnetometer was constructed. Among them, to be closer to the real situation, a combination of the geomagnetism model, the elliptic shell model and the magnetic dipole model was used to simulate the internal magnetic field of the underwater vehicle. The experimental results indicated that the root mean square error of geomagnetic measurement in an underwater vehicle was less than 5 nT after compensation, and the accuracy of geomagnetic measurement met the requirements of geomagnetic matching aided navigation.

## 1. Introduction

With the ongoing development of the world’s economy and military capabilities, underwater tasks are becoming more and more diversified, and the demand for navigation, positioning accuracy and reliability of underwater vehicles (including submarines, unmanned underwater vehicles, self-propelled mines, mines, etc.) is growing day by day. At present, underwater navigation and positioning mainly adopts the inertial navigation system (INS). However, using only pure INS, the positioning error accumulates and diverges with time, making it difficult to ensure that underwater navigation and positioning maintains high precision for a long time. Considering this limitation, INS is usually corrected in real time or periodically by matching aided navigation based on a certain geophysical field. The common methods include terrain matching aided navigation [1], gravity matching aided navigation [2,3] and geomagnetic matching aided navigation [4,5]. The traditional matching algorithms include TERCOM, ICCP, SITAN, matching algorithms based on the Kalman filter and other intelligent matching algorithms. Geomagnetic matching aided navigation has the characteristics of passivity, no radiation, strong concealment and no accumulation of errors over time, and can be used for all-weather and all-regional navigation, which makes it an important research direction for underwater aided navigation in geophysical fields. Using a magnetometer to quickly and accurately measure geomagnetic data is the key element to realizing geomagnetic matching aided navigation. When applying local magnetic matching aided navigation to underwater vehicles, the magnetometer must be installed inside the vehicle, taking into account the navigation safety and concealment of the underwater vehicle [6]. At this time, the total magnetic field obtained by a ship-borne magnetometer is the superposition of the geomagnetic field, carrier magnetic field and random magnetic field. Therefore, the interference magnetic field in underwater carrier geomagnetic measurement mainly comes from the carrier magnetic field (solid magnetic field, induced magnetic field), the random magnetic field and the instrument error of the magnetometer itself, with the carrier magnetic field accounting for a relatively large proportion [7,8,9]. Interference magnetic fields will seriously affect the accuracy of geomagnetic field measurements, thereby affecting the accuracy of geomagnetic matching navigation [10,11,12,13]. Therefore, improving the accuracy of geomagnetic surveys through error compensation has become one of the most urgent and difficult problems to solve in geomagnetic matching navigation.

A lot of research has been conducted on the error compensation method of magnetometers inside vehicles. When the attitude of the vehicle is not considered, the magnetometer error mainly comes from the non-orthogonal error, the scale factor error and the zero offset error of the three-axis sensor. Roberto Alonso and others used the TWO-STEP algorithm to estimate the scale factor and non-orthogonal correction, which has demonstrated good performance [14]. Demoz Gebre Egziabher used the TWO-STEP algorithm and least square estimation to correct the experimental data of a three-axis magnetometer, and the results showed that the residual after correction was very small [15]. Zhang Dewen and others put forward a two-step correction method for the component error of a three-axis magnetic sensor to correct the total magnetic field and the components of the three-axis magnetic sensor [16]. Wang Lihui and others combined the total least square method with the regularization method to propose a truncated total least square method to deal with the ill-conditioned problem of random errors on both sides of the observation equation [17]. The simulation results showed that the method effectively suppressed the ill-conditioned influence in magnetometer measurements. Zhang Qi and others put forward a calibration method for a three-axis magnetic field sensor based on the linearization parameter model and deduced the linearization parameter model between the three-axis measured value of the sensor and the scalar value of the magnetic field [18]. Experimental research shows that this method is a convenient and accurate sensor calibration method. S. Bonnet and others put forward a unified algorithm framework for calculating calibration parameters, such as sensor non-orthogonal error and magnetometer installation error, which was proven to be very effective [19].

When the magnetometer is installed inside the vehicle, the observation error caused by the carrier attitude (hard magnetic error) and the interference field error around the magnetometer (soft magnetic error) should also be considered. Pang Hongfeng systematically studied the error mechanism of a ship-borne geomagnetic vector system, three-axis magnetic sensor correction, misalignment error correction between inertial navigation and a magnetic sensor coordinate system and inertial navigation interference component compensation. The vector system after compensation was tested and applied to regional magnetic anomaly measurement, which verified the advantages of the geomagnetic vector system in magnetic anomaly measurement [20]. Wu Zhitian put forward an integrated method of magnetometer calibration and geomagnetic field compensation. Then, the measurement error of a ship-borne three-axis magnetometer was analyzed, and a parameterized model, including magnetometer calibration error and geomagnetic field measurement error, was established [21]. Finally, the total least squares (TLS) algorithm was used to estimate the parameters of the model, and the experimental results showed that a better compensation effect could be achieved. Zhang Ying and others used truncated total least squares (TTLS) technology to solve the ill-conditioned magnetometer calibration problem, and studied the effectiveness of this method through vehicle experiments. Their results showed that this method can effectively alleviate the ill-conditioned influence in magnetometer calibration and obtain a more stable numerical solution [22]. Because the linearization of the magnetometer observation equation introduces noise to both the system matrix A and data vector B simultaneously, the standard least square solution is not optimal.

Apart from the least squares method, there are other methods that can be used to solve the compensation problem of geomagnetic measurement in underwater vehicles, such as the Kalman filter algorithm, the neural network method, the genetic algorithm and the particle swarm optimization algorithm. In view of the current situation of frequent attitude change during flight, compared with the previous single modeling method, Huan S. and others introduced the eddy current magnetic field for integrated modeling and used the EKF algorithm for compensation [23]. The simulation results showed that this method not only reduces the error amplitude significantly, but also effectively filters out the fluctuation caused by the eddy current magnetic field in the attitude compensation of the aircraft. Yang B. and others adopted a joint estimation iterative algorithm to calibrate the three-axis magnetometer, comparing its performance with that of the EKF algorithm and the nonlinear least squares algorithm [24]. The experimental results showed that the joint estimation iterative algorithm reduced the mean average of the three-axis magnetometer’s measured error from 69.5211 nT to 9.241 nT. Cheng Z studied the calibration method of the magnetometer based on BP neural network, which reduced the influence of model error on calibration accuracy [25]. The numerical simulation results showed that this method can reduce the measurement error of the magnetometer to less than 10 nT. Yu P. and others proposed a neural network compensation method based on regression equations and generalized regression neural networks, which can improve the generalization of the compensation model and weaken the effect of over-fitting [26]. ZHU Xingle and others adopted the Adaptive Multi-Population Genetic Algorithm (AMGA) to calculate a ship’s induced and permanent magnetic parameters [27]. The simulation results indicated that this method has fast computing speed and good stability, with the maximum error of three-component geomagnetic field measurement after compensation being less than 10 nT. Li Ting and others proposed a new magnetic field measurement error compensation method based on the damped particle swarm optimization (DPSO) algorithm. The experimental results showed that this algorithm has good suppression capability to compensate for magnetic field measurement error [28].

Based on previous research experience, this paper proposes a new method to compensate for geomagnetic measurement errors in underwater vehicles based on constrained total least squares. In order to eliminate the magnetic interference of underwater vehicles and improve the geomagnetic measurement accuracy inside underwater vehicles, we established a calculation model of the internal magnetic field of underwater vehicles and the geomagnetic measurement error model of a ship-borne magnetometer. Then, we propose a compensation method for the geomagnetic measurement error of a ship-borne magnetometer based on the constrained total least square method. To verify the effectiveness of the method proposed in this paper, a simulation experiment of geomagnetic measurement and compensation of a ship-borne three-axis magnetometer was constructed. Among them, in order to be closer to the real situation, a combination model of the geomagnetism model based on the IGRF (International Geomagnetic Reference Field) model, the induced magnetic field model simulated by the rotating elliptic shell model and the fixed magnetic field model simulated by the magnetic dipole model was used to simulate the internal magnetic field of the underwater vehicle. In the simulation experiment, firstly, a track in the maneuvering area was designed to obtain the data needed to identify the compensation model parameters P and O. Then, another track in the matching area was designed to verify the geomagnetic measurement compensation accuracy of the compensation model with P and O. The experimental results show that the root mean square error of geomagnetic measurement in the underwater vehicle after compensation is less than 5 nT, and the accuracy of geomagnetic measurement meets the needs of geomagnetic matching aided navigation.

## 2. Modeling of the Underwater Vehicle’s Internal Magnetic Field

Previous studies on the magnetic field of underwater vehicles have mostly focused on estimating the magnetic field outside the underwater vehicle’s hull by establishing ellipsoid, magnetic dipole or mixed models [29,30,31]. Qu Xiaohui and others analyzed the application scope of various magnetic field models [32]. For example, with few sensors, the uniform rotating ellipsoid model is generally used to achieve rapid positioning of the magnetic target. If there are enough sensors near the field source, a mixed model of ellipsoid and magnetic dipole models can be used to improve positioning accuracy. If the sensor is far away from the magnetic target, a single magnetic dipole model can be used. In order to simulate the underwater vehicle’s internal magnetic field more accurately, a combination model was adopted in this paper. This model included the geomagnetism model based on the IGRF (International Geomagnetic Reference Field) model, the induced magnetic field model simulated by the rotating elliptic shell model and the fixed magnetic field model simulated by the magnetic dipole model.

### 2.1. Global Geomagnetic Field Calculation

As an internationally accepted global geomagnetic standard model, the IGRF model describes the Earth’s main magnetic field and its long-term variation. In December 2019, the International Association of Geomagnetism and Aviation (IAGA) released the 13th generation International Geomagnetism Reference Field (IGRF13), which uses the Gaussian spherical harmonic analysis model, based on the reference sphere and the spherical center coordinate system, to represent any point of the global geomagnetic field on the ground or above. According to [33], the scalar potential function V(r,θ,ϕ,t) of the main magnetic field can be expressed as a spherical harmonic series in the IGRF model, as follows:(1)$$V(r,\theta ,\phi ,t)=a{\sum_{n=1}^{N}\sum_{m=0}^{n}\left(\frac{a}{r}\right)}^{n+1}\left[g_{n}^{m}\left(t\right)\cos m\phi +h_{n}^{m}\left(t\right)\sin m\phi \right]P_{n}^{m}\left(\cos\theta \right)$$
where r, θ and ϕ represent the coordinates in the geocentric spherical coordinate system. r represents the radial distance from any point in space to the center of the geomagnetic reference sphere, measured in km and θ, ϕ represent the geocentric latitude and longitude, respectively. a is the radius of the reference sphere, which is close to the average radius of the earth. Usually, we use a=6371.2km. $P_{n}^{m}\left(\cos\theta \right)$ is an associative Legendre function with N-order and M-order Schmidt quasi-normalization. $g_{n}^{m}\left(t\right)$ and $h_{n}^{m}\left(t\right)$ are Gaussian spherical harmonic coefficients with time variation, which can be calculated using IGRF13.

In the passive region of near-Earth space, the main magnetic field caused by the field source inside Earth can be expressed as the negative gradient of the scalar potential function B=−∇V. In the spherical coordinate system, the relationship between components $X_{e}$, $Y_{e}$ and $Z_{e}$ (horizontal component $X_{e}$ in the North, horizontal component $Y_{e}$ in the East, vertical component $Z_{e}$) and the magnetic potential V is as follows:(2)$$X_{e}=\frac{1}{r}\frac{\partial V}{\partial \theta },\;Y_{e}=-\frac{1}{r\sin\theta }\frac{\partial V}{\partial \phi },\;Z_{e}=\frac{\partial V}{\partial r}$$

The geomagnetic components $X_{e}$, $Y_{e}$ and $Z_{e}$ can be calculated using the following Equation:(3)$$\begin{array}{l} X_{e}={\sum_{n=1}^{N}\sum_{m=0}^{n}\left(\frac{R}{r}\right)}^{n+2}\left[g_{n}^{m}\left(t\right)\cos\left(m\phi \right)+h_{n}^{m}\left(t\right)\sin\left(m\phi \right)\right]\frac{d}{d\theta }P_{n}^{m}\left(\cos\theta \right)\\ Y_{e}={\sum_{n=1}^{N}\sum_{m=0}^{n}\left(\frac{R}{r}\right)}^{n+2}\frac{m}{\sin\theta }\left[g_{n}^{m}\left(t\right)\sin\left(m\phi \right)-h_{n}^{m}\left(t\right)\cos\left(m\phi \right)\right]P_{n}^{m}\left(\cos\theta \right)\\ Z_{e}=-{\sum_{n=1}^{N}\sum_{m=0}^{n}\left(\frac{R}{r}\right)}^{n+2}\left(n+1\right)\left[g_{n}^{m}\left(t\right)\cos\left(m\phi \right)+h_{n}^{m}\left(t\right)\sin\left(m\phi \right)\right]P_{n}^{m}\left(\cos\theta \right) \end{array}$$

When the cutoff order N and the Gaussian spherical harmonic coefficient are determined, the three components of the main magnetic field can be calculated using Equation (3).

### 2.2. Calculation of the Internal Magnetic Field of the Rotating Ellipsoidal Shell Model

To analyze the magnetization of the iron rotating ellipsoid shell in a uniform external magnetic field, we should consider the additional magnetic field and the demagnetization coefficient generated by the uniformly magnetized rotating ellipsoid before calculating its internal magnetic field and the magnetic induction intensity.

As shown in Figure 1, the long and short semi-axes of the rotating ellipsoid shell’s outer surface and inner surface are $a_{e}$, $b_{e}$ and $a_{i}$, $b_{i}$, respectively, and their half focal length is g.

According to [34], when the external magnetization field source is uniform, the magnetic field in the cavity of the ellipsoidal shell after magnetization is a uniform field, which is consistent with the direction of the magnetization field source. The shielding coefficients (the ratio of the magnetic field in the cavity after magnetization to the external magnetic field) for magnetization along the longitudinal axis and transverse axis directions are as follows:(4)$$L_{l}=\frac{1}{1+\frac{\chi_{m}}{\mu_{r}}\left[\mu_{r}N_{el}R-\left(N_{il}-KN_{el}\right)\left(1+\chi_{m}N_{el}\right)\right]}$$
(5)$$L_{s}=\frac{1}{1+\frac{\chi_{m}}{\mu_{r}}\left[\mu_{r}N_{es}R-\left(N_{is}-KN_{es}\right)\left(1+\chi_{m}N_{es}\right)\right]}$$
where, $L_{1}$ and $L_{s}$ are the shielding coefficients when magnetized along the longitudinal axis and the transverse axis, respectively. $\mu_{r}$ is the relative permeability of the ferromagnetic shell. $\chi_{m}=\mu_{r}-1$ is the permeability of the ferromagnetic shell. $N_{el}$ and $N_{es}$ are the demagnetization coefficients in the direction of the long and short axes of the solid ellipsoid, respectively, with long and short semi-axes $a_{e}$, $b_{e}$. $N_{il}$ and $N_{is}$ are the demagnetization coefficients in the direction of the long and short axes of the solid ellipsoid, respectively, with long and short semi-axes $a_{i}$, $b_{i}$.

The parameters K and R can be calculated as follows: $K=\frac{a_{i}b_{i}^{2}}{a_{e}b_{e}^{2}}$, R=1−K.

In the ellipsoidal coordinate system, the external magnetic field components in the long axis direction, short axis direction and vertical direction are denoted as $B_{el}$, $B_{es}$ and $B_{ez}$, respectively. According to the shielding coefficient, the magnetic induction intensity components in the rotating ellipsoid shell can be calculated using Equation (6), as follows:(6)$$B_{x}=L_{l}\times B_{el},B_{y}=L_{s}\times B_{es},B_{z}=L_{s}\times B_{ez}$$

### 2.3. Magnetic Dipole Field Calculation

The main components of an underwater ships’ magnetic field are generated not only from the iron pressure-resistant shell magnetized by the geomagnetic field, but also from the internal electrical equipment (such as the propulsion motor, the inertial navigation system, etc.). This part of the magnetic field has a certain relationship with the vehicle’s working state. When the working state of the carrier is stable, the magnetic field caused by the vehicle’s electrical equipment can be regarded as a fixed value [35]. When the interference source is at a certain distance from the magnetic sensor, it can be approximated as a single magnetic dipole. To simplify the problem, a magnetic dipole was used to simulate the total magnetic field generated by all of the internal electrical equipment, based on the superposition principle of magnetic fields. As shown in Figure 2, a right-hand space rectangular coordinate system was established with the center of the magnetic dipole as the origin, where the circular current intensity is I and the radius is R.

The magnetic moment is defined as follows:(7)$${\vec{P}}_{m}=I\cdot S\cdot \vec{n}$$
where S is the area of the plane surrounded by the circular current; that is, $S=\pi R^{2}$. $\vec{n}$ is the normal vector of the plane of the circular current, whose direction conforms to the right-hand rule with respect to I. The Z-axis is in the direction of the magnetic moment vector positively. M is a point in space with spherical coordinates $M\left(r,\varphi_{0},\theta_{0}\right)$, where r is the distance from the coordinate system origin to M, $\varphi_{0}$ is the zenith angle and $\theta_{0}$ is the rotation angle of the meridian plane, where M is located relative to the meridian plane of the X-axis. The conversion relationship between the rectangular coordinates and the spherical coordinates is as follows:(8)$$\left\{\begin{array}{l} x=r\sin\varphi_{0}\cos\theta_{0}\\ y=r\sin\varphi_{0}\sin\theta_{0}\\ z=r\cos\varphi_{0} \end{array}\right.$$
where 0≤r<+∞, $0\leq \varphi_{0}<\pi$, $0\leq \theta_{0}<2\pi$.

Let the magnetic dipole in the space have magnetic permeability μ, intercepting a current element Idl at any point $P\left(R,\pi /2,\theta \right)$ on the circumference. According to Biot–Savart law, the magnetic induction intensity vector generated by this current element at M is as follows:(9)$$d\vec{B}=\frac{\mu }{4\pi }\frac{Id\vec{l}\times \vec{a}}{a^{3}}\begin{array}{c} \left|a\right| \end{array}\neq 0$$
where $d\vec{l}$ is the circumferential tangent vector at P and $\vec{a}$ is the vector PM. Then,
(10)$$d\vec{l}\times \vec{a}=\left|\begin{array}{ccc} \vec{i} & \vec{j} & \vec{k}\\ -R\sin\theta d\theta & R\cos\theta d\theta & 0\\ r\sin\varphi_{0}\cos\theta_{0}-R\cos\theta & r\sin\varphi_{0}\sin\theta_{0}-R\sin\theta & r\cos\varphi_{0} \end{array}\right|$$

Let the magnetic induction intensity be $\vec{B}=B_{x}\vec{i}+B_{y}\vec{j}+B_{z}\vec{k}$. By substituting Equation (10) into Equation (9), we obtain the following:(11)$$\left\{\begin{array}{l} B_{x}=\frac{\mu IR\cdot r\cos\varphi_{0}}{4\pi }\int_{0}^{2\pi }\frac{\cos\theta }{a^{3}}d\theta \\ B_{y}=\frac{\mu IR\cdot r\cos\varphi_{0}}{4\pi }\int_{0}^{2\pi }\frac{\sin\theta }{a^{3}}d\theta \\ B_{z}=\frac{\mu IR}{4\pi }\int_{0}^{2\pi }\frac{R-r\sin\varphi_{0}\cos\left(\theta -\theta_{0}\right)}{a^{3}}d\theta \end{array}\right.$$

Integrating and substituting Equation (11) yields the following:(12)$$\left\{\begin{array}{l} B_{x}=\frac{3\mu }{8\pi }\frac{P_{m}}{{\left(\sqrt{R^{2}+r^{2}}\right)}^{3}}\frac{r^{2}\sin2\varphi_{0}\cos\theta_{0}}{\left(R^{2}+r^{2}\right)}\\ B_{y}=\frac{3\mu }{8\pi }\frac{P_{m}}{{\left(\sqrt{R^{2}+r^{2}}\right)}^{3}}\frac{r^{2}\sin2\varphi_{0}\sin\theta_{0}}{\left(R^{2}+r^{2}\right)}\\ B_{z}=\frac{\mu }{2\pi }\frac{P_{m}}{{\left(\sqrt{R^{2}+r^{2}}\right)}^{3}}\left(1-\frac{3r^{2}\sin^{2}\varphi_{0}}{2\left(R^{2}+r^{2}\right)}\right) \end{array}\right.$$

When the magnetic moment of the dipole is known, the magnetic induction intensity vector at any point in space can be obtained using Equation (12).

## 3. Error Model and Compensation Algorithm of the Ship-Borne Magnetometer

### 3.1. Error Model

The measurement error of the ship-borne magnetometer mainly includes the magnetometer’s instrument error and the underwater vehicle’s interference field error. Instrument error mainly includes three-component non-orthogonal error, scale factor error and zero offset error. Interference field error mainly includes hard magnetic error, soft magnetic error and random error. The error model is as follows:(13)$$B_{SI}^{b}=C_{SI}\left(B_{e}^{b}+B_{HI}^{b}\right)$$
where $B_{e}^{b}$ is the geomagnetic field vector in the volume coordinate system. $B_{HI}^{b}$ is the hard magnetic error vector. $B_{SI}^{b}$ is the soft magnetic error vector. $C_{SI}$ is the soft magnetic coefficient matrix.

In the body coordinate system, the measurement output of the ship-borne magnetometer can be modeled as follows:(14)$$\begin{array}{l} B_{s}=C_{NO}C_{SF}\left(B_{e}^{b}+B_{HI}^{b}+B_{SI}^{b}\right)+b_{o}^{b}\\ C_{NO}=\left[\begin{array}{ccc} 1 & 0 & 0\\ \sin\rho & \cos\rho & 0\\ \sin\varphi \cos\lambda & \sin\lambda & \cos\varphi \cos\lambda \end{array}\right],C_{SF}=\left[\begin{array}{ccc} a & 0 & 0\\ 0 & b & 0\\ 0 & 0 & c \end{array}\right] \end{array}$$
where $B_{s}$ represents the measurement output vector of the ship-borne magnetometer. ρ, φ and λ are the total non-orthogonal error angles and $C_{NO}$ represents the non-orthogonal error matrix. a, b and c are the total scale factor errors and $C_{SF}$ represents the scale factor error matrix. $b_{o}^{b}$ represents the sensor zero bias error vector.

Substituting Equation (13) into Equation (14) yields the following:(15)$$\begin{array}{l} B_{s}=C_{NO}C_{SF}\left[B_{e}^{b}+B_{HI}^{b}+C_{SI}\left(B_{e}^{b}+B_{HI}^{b}\right)\right]+b_{o}^{b}\\ \begin{array}{c} \;\;\;\;=C_{NO}C_{SF}(I_{3\times 3}+C_{SI})(B_{e}^{b}+B_{HI}^{b})+b_{o}^{b} \end{array} \end{array}$$

Simplifying Equation (15) yields the following:(16)$$B_{s}=KB_{e}^{b}+O$$
where $K=C_{NO}C_{SF}\left(I_{3\times 3}+C_{SI}\right)$ and $O=KB_{HI}^{b}+b_{o}^{b}$.The parameter matrices K and O can be calculated as follows [36]:(17)$$K=\frac{1}{\left\|B_{e}\right\|}\left[\begin{array}{ccc} a & 0 & 0\\ b\sin\rho & b\cos\rho & 0\\ c\sin\varphi \cos\lambda & c\sin\lambda & c\cos\varphi \cos\lambda \end{array}\right],O=\left[\begin{array}{c} o_{1}\\ o_{2}\\ o_{3} \end{array}\right]$$
where $o_{1}∼o_{3}$ are the total zero bias errors. $B_{e}$ is the true value of the geomagnetic field.

The total non-orthogonal error angle comes from the manufacturing process error of the magnetometer, the installation error of the magnetometer and the soft magnetic error. The total scale factor error comes from the sensor scale factor error and the soft magnetic error. The total bias error consists of sensor bias error and hard magnetic error. The parameter matrix K is a lower triangular matrix. This is to reduce the model parameters. It is assumed that the X-axis of the sensor coordinate system is completely coincident with the X-axis of the body coordinate system to ensure the uniqueness of K. The nine parameters in Equation (17) describe the input and output process of the three-axis magnetometer completely [37]. It should be noted that the position accuracy of the magnetic field sensor inside the underwater vehicle does indeed have an impact on geomagnetic measurements. Specifically, different installation positions of the sensor can lead to bias errors in geomagnetic measurements. However, this bias error is already included in the **O** matrix of error model and will eventually be eliminated by the error compensation calculation, so there is no need to consider the accuracy of the sensor’s position. However, in the actual measurement process, to avoid introducing significant interference errors, we still tried to keep the sensors away from the magnetic interference sources as much as possible.

The error compensation model of the three-axis magnetometer can be obtained by the inverse calculation of Equation (16), as follows:(18)$${\hat{B}}_{e}^{b}=K^{-1}\left(B_{s}-O\right)=P\left(B_{s}-O\right)$$
where ${\hat{B}}_{e}^{b}$ is the projection of the compensated geomagnetic field measured value in the body coordinate system, and the matrix $P=K^{-1}$ is also a lower triangular matrix. The compensation process of the magnetometer is equivalent to identifying the nine parameters in Equation (17).

### 3.2. Compensation Algorithm Based on Constrained Total Least Squares

It is assumed that the output of the compensated magnetometer $B_{s}$ is equal to the local geomagnetic reference field $B_{e}^{b}$. Then, the following equation holds:(19)$$R={\left\|{\hat{B}}_{b}\right\|}_{2}={\left\|P(B_{s}-O)\right\|}_{2}$$

According to the principle of modal testing, the compensation value is obtained by minimizing the square of difference between the magnetometer’s output and the mode value of the geomagnetic field to realize the error compensation of the geomagnetic field value, as follows:(20)$$\arg\min f(P,O)={\left(R-{\left\|P(B_{s}-O)\right\|}_{2}\right)}^{2}$$

For convenience, we squared both sides of Equation (34) to obtain the following equation:(21)$$R^{2}={\left\|{\hat{B}}_{b}\right\|}_{2}^{2}={\left\|P(B_{s}-O)\right\|}_{2}^{2}$$

The objective function becomes:(22)$$\begin{array}{c} \arg\min f(P,O)={\left(R^{2}-{\left\|P(B_{s}-O)\right\|}_{2}^{2}\right)}^{2}\\ \;\;\;\;\;\;\;\;\;\;\;\;\;\;\;\;\;\;\;\;\;\;\;\;\;\;\;\;\;\;\;\;\;\;\;\;\;\;\;\;\;\;\;\;\;\;\;\;\;\;\;\;\;\;\;\;\;\;\;\;\;={\left(R-{\left\|P(B_{s}-O)\right\|}_{2}\right)}^{2}{\left(R+{\left\|P(B_{s}-O)\right\|}_{2}\right)}^{2} \end{array}$$

Substituting $B_{s}={\left[B_{x}^{s},B_{y}^{s},B_{z}^{s}\right]}^{T}$ and Equation (17) into Equation (22) and expanding, then simplifying it, yields the following model:(23)$$\begin{array}{r} a_{1}(B_{x}^{s}{)}^{2}+a_{2}B_{x}^{s}B_{y}^{s}+a_{3}B_{x}^{s}B_{z}^{s}+a_{4}{\left(B_{y}^{s}\right)}^{2}+a_{5}B_{y}^{s}B_{z}^{s}+a_{6}{\left(B_{z}^{s}\right)}^{2}\\ +a_{7}B_{x}^{s}+a_{8}B_{y}^{s}+a_{9}B_{z}^{s}+a_{10}=0 \end{array}$$
where $a_{i},\begin{array}{c} i=1,2,\cdots 10 \end{array}$ are intermediate variables defined by error compensation parameters P and O. The transformation formula from $a_{i},\begin{array}{c} i=1,2,\cdots 10 \end{array}$ to the parameters of a, b, c and ρ, φ, λ in P, O is as follows:(24)$$\begin{array}{l} X=-\frac{1}{a_{6}}{\left[a_{1},a_{2},a_{3},a_{4},a_{5},a_{7},a_{8},a_{9},a_{10}\right]}^{T}={\left[A,B,C,D,E,G,H,I,J\right]}^{T}\\ \left\{\begin{array}{l} \alpha_{1}=-B^{2}+DC^{2}+4DA+AE^{2}-BEC\\ \alpha_{2}=4AE^{2}J-E^{2}G^{2}-4BECJ+2ECHG+2BEIG-4EHAI-4DICG-C^{2}H^{2}\\ \begin{array}{c} \;\;\;\;+4DAI^{2}+2CBHI-4DG^{2}+4DC^{2}J+4BHG-4AH^{2}-B^{2}I^{2}-4B^{2}J+16DAJ \end{array}\\ \alpha_{3}=E^{4}A-CBE^{3}+E^{2}C^{2}D-2B^{2}E^{2}+8DAE^{2}-4DB^{2}+16D^{2}A \end{array}\right.\\ \left\{\begin{array}{l} \beta_{1}=2BH+BEI-2CDI-4DG+ECH-E^{2}G\\ \beta_{2}=-2AEI+4AH-BCI-2BG+C^{2}H-CEG\\ \beta_{3}=4DIA-2DGC+EGB-IB-2EHA+CBH \end{array}\right.\\ \left\{\begin{array}{l} a=\frac{1}{2\alpha_{1}}\sqrt{-\alpha_{2}(4D+E^{2})}\\ b=\frac{1}{2\alpha_{1}}\sqrt{-\alpha_{2}(4A+C^{2})}\\ c=\frac{1}{2\alpha_{1}}\sqrt{\alpha_{2}(4DA-B^{2})} \end{array}\right.\;\;\;\;\;\;\;\;\;\;\;\;\;\;\;\;\;\;\;\;\;\;\begin{array}{c} \begin{array}{c} \left\{\begin{array}{l} \tan\rho =-\frac{1}{2\sqrt{\alpha_{1}}}(2B+EC)\\ \tan\varphi =\frac{1}{\sqrt{\alpha_{1}}}(BE-2CD)\\ \tan\lambda =E\sqrt{-\alpha_{1}/\alpha_{3}} \end{array}\right. \end{array} \end{array} \end{array}$$

Assuming that the number of geomagnetic field measurements is q, the observation equation composed of these measurements can be combined into a linear equation in matrix form as follows:(25)AX=b
where
  A=Bx1s2Bx1sBy1sBx1sBz1sBy1s2By1sBz1sBx1sBy1sBz1s1Bx2s2Bx2sBy2sBx2sBz2sBy2s2By2sBz2sBx2sBy2sBz2s1⋯⋯      ⋯Bxqs2BxqsByqsBxqsBzqsByqs2ByqsBzqsBxqsByqsBzqs1X=−1a6a1a2a3a4a5a7a8a9a10       Tb=Bz1s2Bz2s2⋯Bzqs2T                                                    

Because q is much larger than the unknown parameters in the observation equation, Equation (25) is an over-determined linear equation set. By solving the unknown parameter vector $\hat{X}$, the estimated $\hat{P}$ and $\hat{O}$ of the compensation parameters can be calculated by substituting Equation (24) into Equation (17).

If the system matrix A is accurate and the data vector b is disturbed by noise, the solution of Equation (25) can be estimated using the standard least square method [38]:(26)$$X_{LS}={\left(A^{T}A\right)}^{-1}A^{T}b$$

However, for the error compensation problem of the magnetometer, the linearization of the observation equation causes both A and b to be disturbed by noise at the same time, resulting in the standard least square solution not being optimal.

We expand the matrices A and b by Taylor series, and their first-order approximate perturbation is:(27)$$\begin{array}{l} \bar{A}=A+\Delta A\\ \bar{b}=b+\Delta b \end{array}$$
where $\bar{A}$ and $\bar{b}$ represent the error-free system matrix and the data vector, respectively. ΔA and Δb are the corresponding perturbation terms, which are related. Then, it is appropriate to solve this problem using the constrained total least squares (CTLS) method [39]. Since Golub and others put forward the concept of TLS [40], this method has been widely used in signal processing, statistics, computers and so on. Based on the mathematical model, we discuss the TLS of Singular Value De-composition (SVD)[41]. CTLS is an extension algorithm of TLS, which reduces the problem to an unconstrained minimization problem on a small variable set. In this paper, the constrained total least squares method, based on the SVD solution, was adopted.

Firstly, the random noise vector is defined as follows:(28)$$v={\left[\begin{array}{cccccccccc} {\tilde{B}}_{x1}^{s} & {\tilde{B}}_{y1}^{s} & {\tilde{B}}_{z1}^{s} & {\tilde{B}}_{x2}^{s} & {\tilde{B}}_{y2}^{s} & {\tilde{B}}_{z2}^{s} & \cdots & {\tilde{B}}_{xq}^{s} & {\tilde{B}}_{yq}^{s} & {\tilde{B}}_{zq}^{s} \end{array}\right]}^{T}$$

According to Equation (28), v∈ℜ3q×1 and v∼ℕ0,σ2I3q, we can transform ΔA and Δb into the following form using random noise vectors, as follows:(29)ΔA=F1vF2v⋯F9v, Δb=F10v

The expressions of $F_{i}\in {ℜ}_{q\times 3q}$, i=1,2,…,10 and $F_{i}$ can be found in [42].

Substituting $\bar{A}$ and $\bar{b}$ into the over-determined Equation (25) to ensure compatibility, we obtained the following:(30)$$\left(A+\Delta A\right)X=b+\Delta b$$

Substituting Equation (29) into Equation (30) and arranging it yielded the following Equation:(31)$$\begin{array}{c} AX-b=-\Delta AX+\Delta b\\ \;\;\;\;\;\;\;\;\;\;\;\;\;\;\;\;\;\;\;\;\;\;\;\;\;\;\;\;\;\;\;\;\;\;\;\;\;\;\;\;\;\;\;\;\;\;\;=-\left[\begin{array}{cccc} F_{1}v & F_{2}v & \cdots & F_{9}v \end{array}\right]X+F_{10}v\\ \;\;\;\;\;\;\;\;\;\;\;\;\;\;\;\;\;\;\;\;\;\;\;\;\;\;\;\;\;=-\left(\sum_{i=1}^{9}x_{i}F_{i}-F_{10}\right)v\\ \;\;=-H_{x}v \end{array}$$
where $x_{i}$ is the *i*-th element of the vector X, $H_{x}=\sum_{i=1}^{9}x_{i}F_{i}-F_{10}$. Thus, the problem of magnetometer error compensation can be transformed into a mathematical problem solved by CTLS, as follows:(32)$$\underset{X,v}{\min}{\left\|v\right\|}_{F}^{2}\;\;\begin{array}{c} s.t. \end{array}\;\;\begin{array}{c} Ax-b+H_{X}v=0 \end{array}$$
where ${\left\|*\right\|}_{F}^{2}$ is a Frobenius norm. According to Equation (32), the random noise vector **v** is estimated as follows:(33)$$\hat{v}=-H_{x}^{†}\left(AX-b\right)$$
where $H_{x}^{†}={\left(H_{x}^{T}H_{x}\right)}^{-1}H_{x}^{T}$ is the generalized inverse of $H_{x}$. The following equation can be obtained by substituting $\hat{v}$ into Equation (32):(34)$$\arg\min F\left(X\right)=v^{T}v=\underset{\begin{array}{cccc} with & respect & to & X \end{array}}{{\left(AX-b\right)}^{T}{\left(H_{x}H_{x}^{T}\right)}^{-1}\left(AX-b\right)}$$

The solution of CTLS can be obtained by minimizing Equation (34) with the Newton iteration method, whose iteration equation is as follows:(35)$$X_{r+1}=X_{r}-H_{r}^{-1}T_{r}$$
where r is the number of iterations. $T_{r}$ and $H_{r}$ represent the gradient vector and Hessian matrix of the r−th iteration, respectively. The specific expressions of $T_{r}$ and $H_{r}$ are shown in [31]. The convergence criterion of the algorithm is to calculate whether the F norm of $T_{r}$ is less than the threshold $\varepsilon_{stop}$; that is, ${\left\|T_{r}\right\|}_{F}<\varepsilon_{stop}$.

The error compensation steps of the three-axis magnetometer based on the CTLS method are as follows:

Step 1: use the SVD solution as the initial value of the Newton iteration method;

Step 2: calculate $X_{r+1}$ according to Equation (35);

Step 3: repeat Step 2 until the following convergence condition is met: ${\left\|T_{r}\right\|}_{F}<\varepsilon_{stop}$;

Step 4: calculate the nine correction parameters $\rho ,\varphi ,\lambda ,a,b,c,o_{1},o_{2},o_{3}$ according to the solution ${\hat{X}}_{r+1}$ of CTLS, and then obtain the estimated compensation matrices $\hat{P}\left(=K^{-1}\right)$ and $\hat{O}$;

Step 5: calculate the compensated geomagnetic measurement value according to Equation (18) with $\hat{P}\left(=K^{-1}\right)$ and $\hat{O}$.

## 4. Experiment

### 4.1. Output Simulation of the Magnetometer

The simulation area was set to be in the range of 120°~120.1° East longitude and 25.5°~25.6° North latitude, as shown in Figure 3. The regional geomagnetic field was calculated using the IGRF model. In the maneuvering area and the matching area, two tracks of the vehicle were designed to be in the shape of “∞” and a “double circle”, respectively, as shown in Figure 4. The combination of the geomagnetism model, the elliptic shell model and the magnetic dipole model with Gaussian white noise were adopted to simulate the internal magnetic field of the vehicle.

The model parameters of the rotation ellipsoid shell included the major axis (a=51.5 m), the minor axis (b=5.25 m), the thickness of the shell (0.035 m) and the relative permeability ($\mu_{r}=50$).

The parameter of the magnetic dipole model was the magnetic moment $P_{m}=314.6\;A/m^{2}$, and its direction was consistent with the long axis direction of the ellipsoid shell.

Gaussian white noise: $\mu_{0}=0.0$.

The simulation results are shown in Figure 3 and Figure 5. The variation of the geomagnetic field within the vehicle’s track was about 40 nT, whereas the variation of the additional magnetic field generated by the magnetization of the vehicle’s iron ellipsoid shell was more than 10000 nT without the influence of the magnetic dipole interference field and random noise. Therefore, it is impossible to obtain the real geomagnetic field value without error compensation.

### 4.2. Compensation Simulation of the Magnetometer’s Measurement

In order to test the robustness of the algorithm in this paper, we added Gaussian white noise to the simulation data in the maneuvering area and set the signal-to-noise ratio (SNR) to 80, 90 and 95. When the vehicle was in the maneuvering area, we obtained the simulation data $B_{e}$ and $B_{s}$. On this basis, the mediate variables $a_{i},\begin{array}{c} i=1,2,\cdots 10 \end{array}$ were identified using the CTLS solution from Equation (25). Then, according to Equation (24), the parameters of a, b, c and ρ, φ, λ were calculated using the mediate variables. Finally, we obtained the compensation parameter matrices **P** and **O**, according to Equation (17). The calculation results and the RMSEs are shown in Figure 6 and Table 1. It can be seen that the RMSE decreases as SNR increases, with a maximum value of less than 3.1 nT. The calculated compensation parameter matrices P and O were used to correct the magnetic field in the matching area.

With the compensation parameter matrices P and O, we calculated the compensated geomagnetic field valve in the matching area (the SNR was also set to 80, 90 and 95). The compensation results are shown in Figure 7.

From the above compensation results, with an increase in SNR from 80 to 90, the compensation error decreased and the RMSE decreased from 2.82 to 0.95. Moreover, we can see that the compensation residual reflects the signal noise level and the uneven field in the region. Therefore, to reduce the compensation error, we should make the vehicle’s moving area as small as possible in the compensation area and try to keep the course unchanged and stable. This will prevent estimation errors caused by the uneven magnetic field in the region.

## 5. Conclusions

The measurement error of a three-axis magnetometer can be described by nine parameters. After establishing the error compensation model, the model can be linearized with the help of intermediate parameters, and then compensation parameters can be quickly obtained using the linear optimization method.

The simulation experiment results show that when the magnetometer is installed inside the underwater vehicle, the observation equation is seriously ill-conditioned due to the influence of iron shell’s demagnetization field and noise. Both the data matrices and parameter matrices contain significant errors, necessitating the use of an improved constrained total least squares method to solve them.

Although the linear solution speed is fast, when the observation data is noisy and the error compensation model is severely ill-conditioned, the CTLS method based on SVD decomposition is biased. Then, the gradient $T_{r}$ in the Gaussian–Newton method increases sharply, and the decrease during iteration is not significant, and may lead to non-converge. It is necessary to improve relevant algorithms to achieve higher compensation accuracy in linear solving.

## Figures and Tables

**Figure 1 sensors-24-03478-f001:**
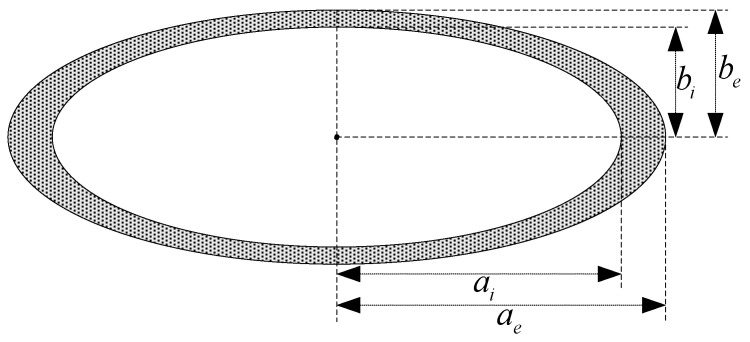
Rotating ellipsoid shell.

**Figure 2 sensors-24-03478-f002:**
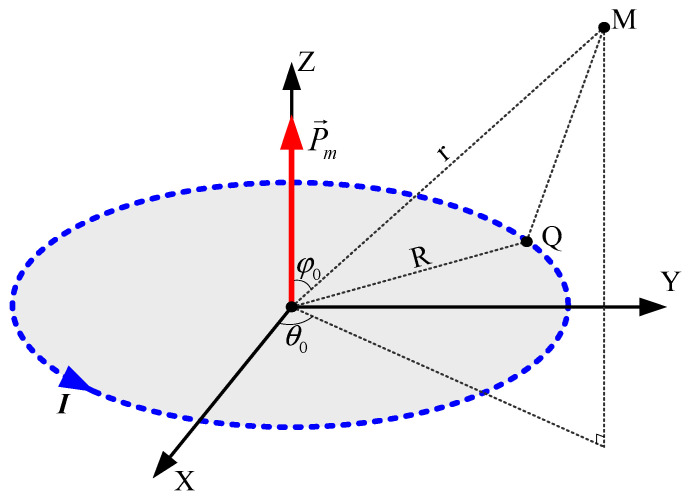
Magnetic dipole in a space rectangular coordinate system.

**Figure 3 sensors-24-03478-f003:**
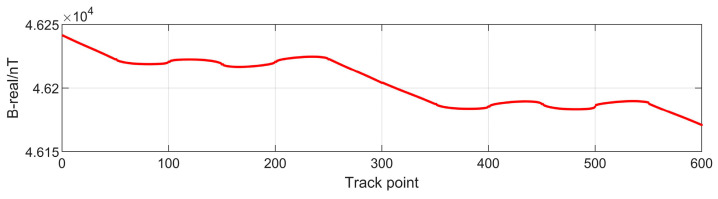
Geomagnetic field data on track points.

**Figure 4 sensors-24-03478-f004:**
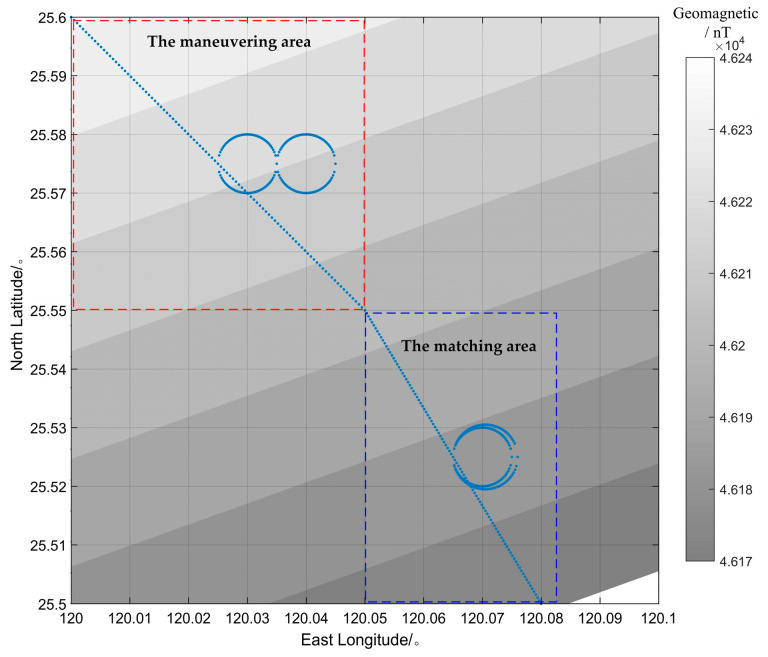
Regional geomagnetic field and the vehicle’s track.

**Figure 5 sensors-24-03478-f005:**
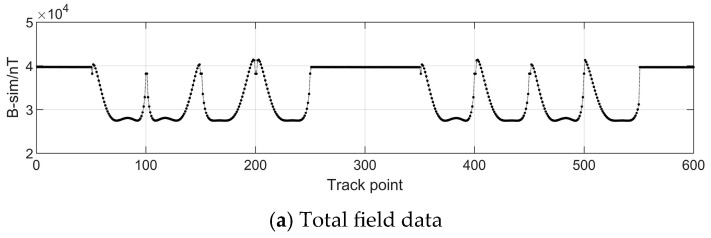
Magnetic field data at track points using the magnetometer’s measurements.

**Figure 6 sensors-24-03478-f006:**
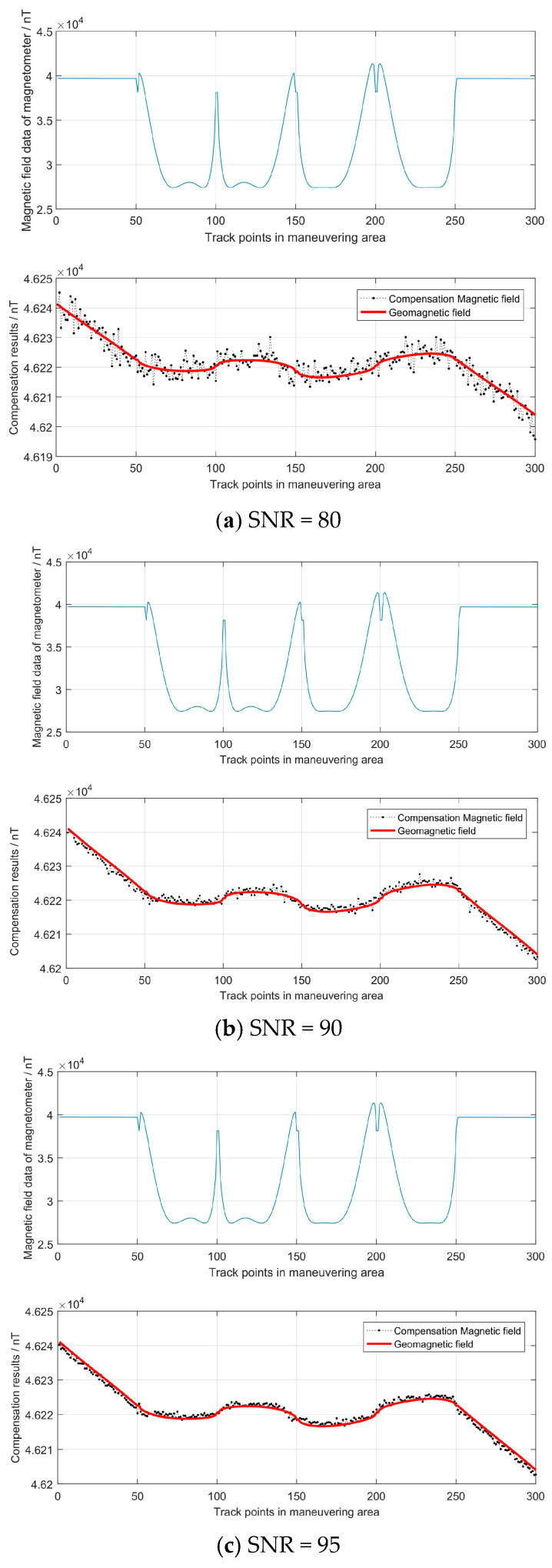
Compensation results in the maneuvering area.

**Figure 7 sensors-24-03478-f007:**
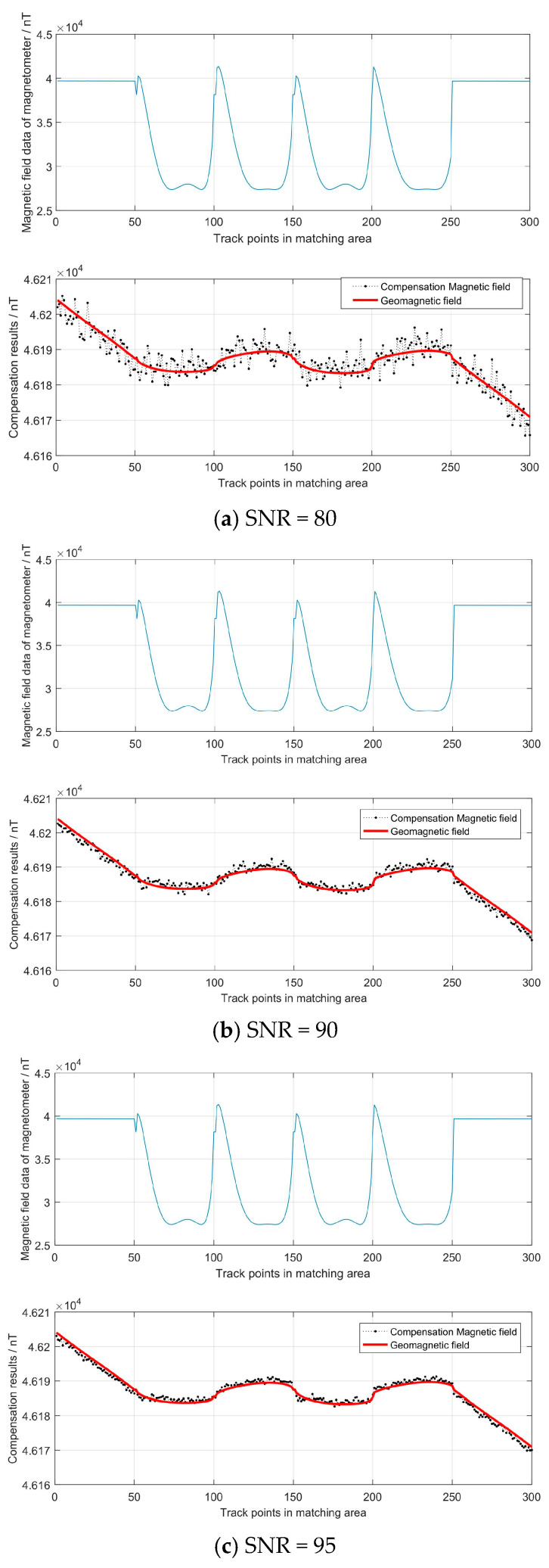
Compensation results in the matching area.

**Table 1 sensors-24-03478-t001:** Magnetometer compensation parameters and the RMSE of compensation error in the maneuvering area.

SNR	P	O	RMSE
**80**	$\left[\begin{array}{ccc} 1.0594 & 0 & 0\\ 0.000946 & 1.688 & 0\\ 0.000414 & 0.000157 & \;1.6875 \end{array}\right]$	$\left[\begin{array}{c} 102.17\\ 21.97\\ 2.84 \end{array}\right]$	3.02
**90**	$\left[\begin{array}{ccc} 1.0598 & 0 & 0\\ 0.001194 & 1.6884 & 0\\ 0.000580 & 0.000306 & 1.6876 \end{array}\right]$	$\left[\begin{array}{c} 117.58\\ 28.19\\ 4.31 \end{array}\right]$	1.86
**95**	$\left[\begin{array}{ccc} 1.0599 & 0 & 0\\ 0.001281 & 1.6885 & 0\\ 0.000660 & 0.000380 & 1.6877 \end{array}\right]$	$\left[\begin{array}{c} 123.65\\ 30.61\\ 5.06 \end{array}\right]$	1.73

## Data Availability

The original contributions presented in this study are included in this article. Further inquiries can be directed to the corresponding author.

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
