# Peer review of "A Compensation Method for the Geomagnetic Measurement Error of an Underwater Ship-Borne Magnetometer Based on Constrained Total Least Squares"

_sensors, 2024, doi:10.3390/s24113478_

Round 1
Reviewer 1 Report
Comments and Suggestions for Authors
The manuscript introduces a so-called Constrained Total Least Squares (CTLS) method to correct magnetic measurement errors in underwater navigation systems. This is derived from a previous method, TLS, from ref. 29. The method minimizes the error with respect to the internationally accepted geomagnetic standard model (IGRF). The authors test and benchmark their method against simulated data.
The methodology seems solid. I am not familiar with the predecessor, TLS, but I did some consistency checks on the maths and they seem correct. What is beyond the scope of the work however, is a test of the algorithm in a real-world scenario, so the implementation remains to be seen.
The only request I would have for publication is that it would be helpful to discuss a bit more the context and integration of the algorithm with other methods typically used to improve inertial navigation, such as filtering techniques.
Comments on the Quality of English Languagejust some minor errors, eg "to solver->to solve" in line 18, "proved->proven" in l 70,...
Author Response
Dear Reviewer,
Please see the attachment.
Yours sincerely,
All the authors

Reviewer 2 Report
Comments and Suggestions for Authors
Kind of mechanic, a lot of English errors in the paper. The authors should examine and eliminate these errors before submitting the paper again.
Author Response

(The authors gave the same response as above.)

Reviewer 3 Report
Comments and Suggestions for Authors
The authors propose a method of geomagnetic measurement error compensation of underwater shipborne magnetometer based on constrained total least square method. A simulation model is constructed on the basis of comprehensive consideration of ship demagnetization field, the experimental results are that the geomagnetic measurement error is less than 5 nT after error compensation. The issue studied in the article is of practical significance. However, there are several problems with this article. Modifications are recommended as follows:
1、the usage of symbols for functions, variables, constant is extremely confusing, making it difficult to understand the content. For example , the function symbols "sh" and "ch" should be written in upright form.
2、the format of the figures and tables in the text is not standardized, which complicates the interpretation of the experimental results.
3、There are few references from the past five years, making it difficult to judge whether a measurement error of 5 nT is advanced.
Comments on the Quality of English LanguageThe writing of the article is acceptable, but the usage of symbols is quite confusing, making it difficult to understand.
Additionally, there are some typographical errors in the article. For example, "To solver this problem" in the abstract. "Magnetic dipole 3and its coordinate system" in the caption of Figure 3.
Author Response

(The authors gave the same response as above.)

Reviewer 4 Report
Comments and Suggestions for Authors
In this manuscript, the authors present a methodology for magnetic compensation based on the tri-axial field measurements inside the underwater carrier. To this end, they first analyze the measurement error of the onboard magnetic field sensors and the geomagnetic measurement error compensation based on constrained total least square method. Moreover, the authors present simulation results concerning the ship demagnetization field, demonstrating the resulting geomagnetic measurement error compensation and thus, the validity of their method. The paper is in general interesting, well-written, very informative and comprehensive. Some minor comments:
In my view, the abstract should be re-written to better illustrate the methodology and results of the paper.
You should include in the introduction one or two paragraphs that summarize the innovation points of your paper.
In Figures 5, 6 and 7, please insert axes units.
Please also mention other methods (apart from least squares) that can be used to solve this problem.
Moreover, one of the main parameters that introduce uncertainty in your modeling process is the accuracy in the position vector of the magnetic field sensors. In this context, it would be interesting to at least discuss this as possible extension/enhancement of your developed methodology. Relevant references:
1. "Calibration of room temperature magnetic sensor array for biomagnetic measurement." IEEE Transactions on Magnetics 55.7 (2019): 1-6.
2. "A software-based calibration technique for characterizing the magnetic signature of EUTs in measuring facilities." IEEE Transactions on Electromagnetic Compatibility 59.2 (2016): 334-341.
3. "Magnetic field sensor calibration for attitude determination." Journal of Applied Geodesy 8.2 (2014): 97-108.
4. "The calibration of 3-axis magnetic sensor array system for tracking wireless capsule endoscope." 2006 IEEE/RSJ International Conference on Intelligent Robots and Systems. IEEE, 2006.
Comments on the Quality of English Language· Small typos are present (e.g., lines 54, 57). Please carefully proofread your manuscript.
Author Response

(The authors gave the same response as above.)

Round 2
Reviewer 3 Report
Comments and Suggestions for Authors
The author provided satisfactory answers to all my questions and made numerous revisions to the article, significantly improving its quality. I recommend accepting the article.